# Assessing the chemical profile and biological potentials of *Tamarix smyrnensis* flower extracts using different solvents by in vitro, in silico, and network methodologies

Erdi Can Aytar[1]*, Emine Incilay Torunoğlu[2]*, Abidin Gümrükçüoğlu[3], Saleh Al-Farraj[4], Alper Durmaz[5], Mika Sillanpää[6,7,8]*

1 Usak University Faculty of Agriculture Department of Horticulture, Uşak, Türkiye, 2 Necmettin Erbakan University, Faculty of Medicine, Department of Medical Biochemistry, Konya, Türkiye, 3 Artvin Çoruh University, Medicinal-Aromatic Plants Application and Research Center, Artvin, Türkiye, 4 King Saud University, College of Science, Department of Zoology, Riyadh, Saudi Arabia, 5 Artvin Çoruh University, Ali Nihat Gökiğit Botanical Garden Application and Research Center, Artvin, Türkiye, 6 Saveetha University, Saveetha School of Engineering, Saveetha Institute of Medical and Technical Sciences, Chennai, Tamil Nadu, India, 7 University of South Africa, College of Science, Engineering and Technology, Institute for Nanotechnology and Water Sustainability (iNanoWS), Florida Campus, Johannesburg, South Africa, 8 Chandigarh University, University Centre for Research & Development, Department of Civil Engineering, Gharuan, Mohali, Punjab, India

* erdicanaytar@gmail.com (EDA); mikaesillanpaa@gmail.com (MS)

## Abstract

This study evaluated the antioxidant properties, phytochemical content, and protein interactions of *Tamarix smyrnensis* flower extracts prepared with different solvents (methanol, ethanol, ethyl acetate). The antioxidant activities were assessed using the DPPH, metal chelation, and CUPRAC assays. Ethanol extracts exhibited the strongest radical scavenging activity ($IC_{50}$=25.15 µg/mL), while methanol extracts showed superior metal chelation (49.84 mg/mL) and reducing power ($IC_{50}$=35.95 µg/mL). HPLC analysis identified key bioactive compounds, with p-coumaric acid, catechin, and gallic acid being the most abundant phenolics and flavonoids across the extracts. Methanol extracts were rich in 4-hydroxybenzoic acid (2253.9 mg/L) and catechin (6974.4 mg/L). Molecular interaction predictions using the STITCH database revealed that phenolic compounds such as 4-hydroxybenzoic acid, catechin, and p-coumaric acid have significant interactions with proteins involved in antioxidant defence, including COQ2 and HMOX1. STRING-based protein–protein interaction (PPI) analysis further highlighted a network of proteins involved in mitochondrial function and oxidative stress response, suggesting that *T. smyrnensis* phenolics could modulate these pathways. Molecular docking studies confirmed that chlorogenic acid and catechin showed strong binding affinities with the antioxidant-related protein PTGS2, indicating their potential as therapeutic agents. The findings emphasise the significant antioxidant potential of *T. smyrnensis* flower extracts, driven by their rich phytochemical content and promising bioactivity, offering a foundation for future therapeutic applications targeting oxidative stress-related diseases.

**Data availability statement:** All data generated or analyzed during this study are presented only in Supplementary Material S1.

**Funding:** This project was supported by Researchers Supporting Project Number (RSP-2026R7) King Saud University, Riyadh, Saudi Arabia. The King Saud University (KSU) funding, this support was requested by KSU and covers the time Prof. Saleh dedicated to working on the manuscript.

**Competing interests:** The authors have declared that no competing interests exist.

## Introduction

*Tamarix smyrnensis* Bunge in Tent. Gen. Tamar.: 53 (1852) belongs to the family Tamaricaceae and is naturally distributed across the Mediterranean Basin and Western Asia. The native range of the species spans from Pakistan in the east to Bulgaria in the west, from southern European Russia in the north to the Lebanon-Syria region in the south, placing Türkiye near the centre of its distribution. It is recognised as one of the most widespread *Tamarix* species within Türkiye, with a presence across all geographic regions [1]. Taxonomically, *T. smyrnensis* has a complex nomenclatural history with at least eight recorded synonyms [1]. It is classified under subclass *Magnoliidae*, within the order *Caryophyllales*. The Tamaricaceae family consists of four accepted genera: *Tamarix*, *Myricaria*, *Myrtama*, and *Reaumuria* [2]., with *Tamarix* being the most speciose, containing 73 accepted species globally [3].

Despite the global presence of 73 *Tamarix* species, only eight species are native to Türkiye, representing approximately 11% of the genus's worldwide diversity. Among these, *Tamarix duezenlii* is endemic to Türkiye and has been identified as a conservation priority due to its restricted distribution [1,3–5]. The focus of this study, *T. smyrnensis*, thrives in moist sandy habitats, typically along riverbanks, and can grow at elevations ranging from sea level up to 1000 meters. The type specimen was collected in 1827 from wet sandy areas near Smyrna (modern-day İzmir, Türkiye) by Fleischer, which also inspired the species [4].

Plant-derived antioxidant compounds have been intensively studied in recent years due to their role in preventing oxidative damage caused by free radicals. Reactive oxygen species (ROS) formed at the cellular level led to damage in DNA, lipids, and proteins, serving as triggers for various pathologies such as chronic inflammation, cardiovascular diseases, and neurodegenerative disorders. Therefore, the antioxidant capacity of phenolic and flavonoid compounds derived from plants has become an increasingly important area of interest in both the food and pharmaceutical industries [6,7].

The antioxidant activity of plant extracts largely depends on the polarity of the solvent used for extraction. High-polarity solvents, such as methanol and ethanol, typically facilitate the extraction of phenolic compounds with high yields. In contrast, moderately polar solvents, such as ethyl acetate, may yield more specific flavonoid profiles [8,9]. Recent studies comparing these solvents through various in vitro antioxidant assays, such as the DPPH, metal chelation, and CUPRAC assays, provide detailed information on the radical scavenging capacity and reducing power of the extracts. Additionally, the quantitative separation of flavonoids and phenolic acids is performed using high-performance liquid chromatography (HPLC) methods, allowing for the sensitive detection of a wide range of compounds, from ascorbic acid to resveratrol [10].

Although various phytochemical studies have been conducted on *Tamarix* species, this is the first comprehensive investigation specifically focusing on the phytochemical composition and antioxidant activity of *T. smyrnensis* flowers. Using a multi-solvent extraction approach combined with DPPH, CUPRAC, and metal chelation assays, along with detailed HPLC-DAD analysis, we reveal the remarkable bioactive

potential of the floral tissues. Unlike previous studies that examined vegetative parts or whole plants, our flower-targeted methodology provides novel insight into the species' antioxidant richness, positioning *T. smyrnensis* flowers as a promising natural source for functional applications. Notably, the inclusion of *in silico* molecular interaction studies and network analysis further strengthens the novelty of this research and clearly distinguishes it from earlier reports in the literature.

In recent years, bioinformatics approaches using databases such as STITCH and STRING have become increasingly common for predicting compound–target protein interactions. For example, phenolics like 4-hydroxybenzoic acid and catechin show high binding scores with targets associated with oxidative stress response and mitochondrial function, such as COQ2, HMOX1, and PTGS2, shedding light on the potential cellular mechanisms of plant antioxidants [11]. These in silico predictions are further validated through molecular docking studies (e.g., PTGS2–chlorogenic acid interaction) and provide valuable insights for the identification of new therapeutic candidates [12].

Building on these methodologies, the present study systematically compared the antioxidant capacities of *T. smyrnensis* extracts obtained using methanol, ethanol, and ethyl acetate via DPPH, metal chelation, and CUPRAC assays. It also quantifies key vitamins, phenolics, and flavonoids by HPLC and employs STITCH- and STRING-based network analysis to identify major protein targets. Finally, molecular docking against HMOX1, PTGS2, and PON1 was performed to assess binding affinities, ligand efficiencies, and interaction patterns, thereby providing an integrated phytochemical–pharmacological profile of *T. smyrnensis*.

## Materials and methods

### Plant collections

Plants belonging to *T. smyrnensis* were collected from homogeneous populations located in marshy areas within the basin of the Yeşilırmak River, close to the Samsun–Çarşamba Fener area in Çarşamba district, Samsun province, Türkiye [13]. Fieldwork was carried out on May 18, 2023, during which flowering branches were sampled for identification purposes. Additionally, flowers were collected for phytochemical content analysis and extraction studies.

To ensure the ecological and chemical integrity of the plant material, samples of *T. smyrnensis* were collected from a geographically isolated population characterised by a homogeneous composition and minimal anthropogenic disturbance. This specific site was selected following repeated field surveys, and sampling was conducted during the peak flowering period to obtain mature generative organs. The restricted collection window was necessary due to the focus on floral tissues, which are only available for a limited time under optimal physiological conditions. This approach allowed for the most accurate representation of the species' generative phase in its natural habitat.

The identification of the species was carried out by Dr. Alper Durmaz, based on the descriptions in *Flora of Turkey and the East Aegean Islands*. To confirm the current taxonomic status and nomenclature, the Plants of the World Online (POWO) database, the "BizimBitkiler" database, and recent literature were consulted. The herbarium specimens of *T. smyrnensis* have been preserved in the Herbarium of the Department of Biology, Faculty of Science, Ondokuz Mayıs University, under the accession number OMUB-8223.

### Plant extraction

Freshly harvested *T. smyrnensis* flowers were carefully cleaned to remove any foreign materials and then dried in an oven at 40 °C for two days. After drying, the plant material was ground into a fine powder using a laboratory mill to ensure a homogeneous extraction process. A total of 100 grams of the powdered sample was weighed and mixed with methanol, ethanol, and ethyl acetate in a 1:10 (w/v) ratio. To prevent photodegradation of sensitive compounds, the mixture was kept in a dark environment for 72 hours under static maceration conditions. After the extraction period, the mixture was filtered to separate the solid and liquid phases. The resulting extracts were concentrated by evaporating the solvent under reduced pressure at 40 °C using a rotary evaporator. The concentrated extract was then transferred into airtight containers and stored at 4 °C for further analyses [14].

*Calculation of Extraction Yield (%)*

$$\text{Extraction yield } (\%) = \text{Initial plant material mass } (g) / \text{Mass of obtained extract } (g) \times 100$$

## Antioxidant analysis

**2,2-Diphenyl-1-Picrylhydrazyl assay.** The antioxidant activity of the plant extracts prepared with methanol, ethanol, and ethyl acetate was assessed using the DPPH assay. For each extract, 0.1 mL of the solution was added to 3.9 mL of freshly prepared DPPH solution. The mixture was gently vortexed and incubated in the dark at room temperature to prevent photodegradation of the radical. After the reaction time had elapsed, the absorbance was measured at 517 nm using a spectrophotometer. The degree of discolouration indicated the scavenging capacity of each extract against DPPH radicals [15].

**CUPRAC reducing power assay.** To determine the CUPRAC ion-reducing antioxidant capacity, 0.1 mL of each extract (methanol, ethanol, and ethyl acetate) was mixed with 1.0 mL of distilled water. Subsequently, copper (II) chloride, neocuproine, and ammonium acetate buffer (pH 7.0) were added in equal volumes (1:1:1 ratio) to obtain a final reaction volume of 4.1 mL. The solution was incubated at room temperature for 30 minutes, allowing complex formation. Absorbance was then recorded at 450 nm. The antioxidant capacity of each extract was expressed as µmol Trolox equivalents per gram of dry plant material [16].

**Ferrous ion chelating activity assay.** The ability of methanol, ethanol, and ethyl acetate extracts to chelate ferrous ions was assessed following a modified version of the method described by Dinis et al. [17] For each assay, 800 µL of the extract (at varying concentrations) was mixed with 10 µL of 0.6 mM $FeCl_2$ solution. The mixture was vortexed and allowed to stand at room temperature for 10 minutes to facilitate interaction. Following this incubation, 50 µL of 5 mM ferrozine solution was added to initiate the chelation reaction. The total volume was adjusted to 1 mL using distilled water. After an additional 10-minute period at room temperature, the absorbance was measured at 562 nm using a spectrophotometer. The percentage of ferrous ion chelation was calculated, and the concentration of extract required to achieve 50% inhibition ($IC_{50}$) was determined by plotting the percentage of metal chelation versus extract concentration.

## Optimised HPLC-DAD method for phenolic compound identification and quantification

Dry powdered samples (1 g each) were extracted separately using 10 mL of three different solvents: methanol, ethanol, and ethyl acetate. To facilitate the efficient release of target phytochemicals into each solvent, the mixtures were first subjected to 30 minutes of ultrasonication. Following this, they were placed on a shaker and incubated in darkness at room temperature for 24 hours to ensure exhaustive extraction. After incubation, the mixtures were filtered using standard filter paper to remove coarse particles. The resulting filtrates were then further clarified using 0.45 µm syringe filters, yielding clear solutions ready for chromatographic analysis. Each extract was analysed using an ACE 5 C18 column (250 mm × 4.6 mm, 5 µm particle size) on an HPLC system. The mobile phase consisted of acetonitrile (Solvent A) and 1.5% acetic acid in water (Solvent B), delivered in a gradient mode starting from 15% A and 85% B and progressing to 40% A and 60% B over 29 minutes. Detection was carried out using a 1260 DAD WR detector at 250 nm, 270 nm, and 320 nm. The system included a 1260 Quaternary Pump (flow rate: 0.7 mL/min), a 1260 Vial Sampler (10 µL injection volume), and a G7116A column oven maintained at 35 °C. Calibration curves were prepared using six standard concentrations (25, 50, 75, 100, 200, and 300 µg/mL) to quantify phenolic compounds in each extract. This multi-solvent approach, combining sonication, extended incubation, precise filtration, and robust HPLC-DAD detection, enabled a comprehensive assessment of solvent-specific extraction efficiency and phenolic profiles [18].

## Network-based interaction prediction

In this study, a computational approach was employed to predict the potential interactions between selected plant-derived phenolic compounds and human proteins. Initially, five phenolic compounds—4-hydroxybenzoic acid, p-coumaric acid, chlorogenic acid, and catechin were converted into SMILES format and submitted to the STITCH database (http://stitch.embl.de/) [19] to obtain compound–protein interaction predictions. The resulting interaction data, including compound names, protein identifiers, and interaction scores, were further analysed using the STRING database (https://string-db.org/) [20] to construct a protein–protein interaction (PPI) network. The STRING output was exported in CSV format and imported into the Cytoscape software [21] for network visualisation and topological analysis. Parameters such as interaction confidence scores, node connectivity, and centrality were used to identify key proteins within the network. Functional modules and biological relevance of the clustered proteins were also evaluated.

## Molecular docking

Three protein targets identified through network-based interaction analysis—HMOX1 (PDB ID: 1N45), PTGS2/COX-2 (PDB ID: 5IKQ), and PON1 (PDB ID: 1V04)—were selected for molecular docking studies. The crystal structures of these proteins were retrieved from the Protein Data Bank (PDB). Prior to docking, water molecules and non-standard ligands were removed, polar hydrogens were added, and Gasteiger charges were assigned using AutoDock Tools (v1.5.7). The grid box was defined to cover the known or predicted active site based on literature or inhibitor-bound co-crystallised ligands. The 3D structures of selected phenolic compounds (4-hydroxybenzoic acid, p-coumaric acid, chlorogenic acid, and catechin) were obtained from the PubChem database in SDF format and converted to PDBQT format using Open Babel and AutoDock Tools. All ligands were energy-minimised using the MMFF94 force field prior to docking. Molecular docking simulations were conducted using AutoDock Vina, and the best binding pose was selected based on the lowest binding energy ($\Delta G$). Binding interactions were visualised and analysed using Discovery Studio Visualizer and BIOVIA tools [22].

## Statistical analysis

Correlation coefficients (R) were calculated using the CORREL statistical function in MS Excel to assess the relationship between the two variables. The data are presented as mean ± SD based on three independent observations. The data analysis was performed using SPSS version 21.

To assess whether the amounts of individual compounds and antioxidant activities differed significantly between the different solvent extracts (methanol, ethanol, and ethyl acetate), we applied one-way ANOVA (analysis of variance). When ANOVA results showed a statistically significant difference ($p < 0.05$), we followed up with a Tukey HSD (Honestly Significant Difference) test to compare the means between the groups and determine which solvent caused the difference.

All statistical analyses were performed using the SPSS Statistics 22.0 software package. The data were expressed as mean ± standard deviation, and comparisons between groups were illustrated using different lowercase letters (e.g., *a*, *b*, *c*)—values sharing the same letter were not statistically different from each other. In contrast, different letters indicated a significant difference.

## Results and discussion

### Extraction yield of *Tamarix smyrnensis*

The extraction yield was determined as the percentage of the obtained dry extract relative to the initial plant material. In this study, 100 g of plant material was extracted with 1000 mL of solvent (ethanol, methanol, or ethyl acetate), resulting in different amounts of dry extract depending on the solvent used. 16 g of dry extract was obtained with ethanol, 18 g with methanol, and 10 g with ethyl acetate, corresponding to extraction yields of 16%, 18%, and 10%, respectively.

## Comparative antioxidant activities of extracts obtained with different solvents

In this study, the antioxidant activity of *T. smyrnensis* flowers was evaluated using DPPH, CUPRAC, and iron chelation activity tests (Table 1). The DPPH radical scavenging capacity of *T. smyrnensis* flower extracts had an $IC_{50}$ value ranging from $27.81 \pm 1.91$ to $65.88 \pm 6.01$ μg/mL. The highest activity was observed in the ethanol extract, with an $IC_{50}$ value of $25.15 \pm 2.07$ μg/mL, followed by the methanol extract with an $IC_{50}$ value of $27.81 \pm 1.91$ μg/mL. The ethyl acetate extract exhibited the lowest antioxidant activity, with an $IC_{50}$ value of $65.88 \pm 6.01$ μg/mL. The CUPRAC radical scavenging activity of *T. smyrnensis* flower extracts ranged from $IC_{50}$ values of $35.95 \pm 2.23$ to $151.46 \pm 3.56$ μg/mL. Among these, the ethanol extract demonstrated the highest activity with an $IC_{50}$ value of $35.95 \pm 2.23$ μg/mL. At the same time, the remaining extracts exhibited comparatively lower activities. Therefore, the extracts of both *Tamarix* species from the flowers exhibited strong anti-radical activity, especially in the DPPH and CUPRAC radical scavenging tests. A similar trend was also observed in the metal chelation tests of the extracts. The $Fe^{3+}$ binding activity of *T. smyrnensis* flower extracts ranged from $IC_{50}$ values of $49.84 \pm 2.83$ mg/mL to $174.12 \pm 4.55$ mg/mL, with the highest effect observed in methanol and ethanol extracts. The $Fe^{3+}$ binding activity of *T. smyrnensis* flower extracts was lowest in the ethyl acetate extract ($IC_{50}$ value of $174.12 \pm 4.55$ mg/mL). The highest binding potential was observed in methanol, which was 1.6 times and 3.5 times higher than ethanol and ethyl acetate extracts, respectively. The *T. smyrnensis* flower extracts exhibited the highest antioxidant activity in the DPPH assay, as well as in the CUPRAC and ethyl acetate assays for methanol extracts. All experimental measurements were performed in triplicate, and the corresponding data are provided in S1 Table.

The polar extract of *Tamarix gallica* has been reported to exhibit strong antioxidant activity in the DPPH assay, with an $IC_{50}$ value of 6 μg/mL [23]; in contrast, in the present study, the ethanol extract of *T. smyrnensis* flowers displayed the highest DPPH radical scavenging activity ($IC_{50} = 25.15 \pm 2.07$ μg/mL), indicating a comparatively lower antioxidant potential than the *T. gallica* extract. Similarly, the ethanol extract of the aerial part of *T. gallica* demonstrated higher antioxidant activity in the DPPH assay compared to the methanol extract, with $IC_{50}$ values of $1.309 \pm 0.31$ mg/mL and $1.606 \pm 0.43$ mg/mL, respectively [24], whereas in our study, the ethanol extract of *T. smyrnensis* flowers exhibited greater antioxidant activity than the methanol extract, aligning with these literature findings.

On the other hand, the ethyl acetate extract of *Tamarix africana* showed a CUPRAC $A_{0.5}$ value of $11.13 \pm 2.65$ μg/mL, while the methanol extract was reported as $13.94 \pm 0.72$ μg/mL, and in the DPPH assay, the $IC_{50}$ values were $5.68 \pm 0.06$ μg/mL and $7.80 \pm 0.11$ μg/mL for the ethyl acetate and methanol extracts, respectively [25]; in our study, the ethyl acetate extract of *T. smyrnensis* flowers exhibited a CUPRAC activity of $151.46 \pm 3.56$, while the methanol extract showed an $IC_{50}$ value of $35.95 \pm 2.23$. These results indicate that *T. africana* exhibited higher activity in the DPPH and CUPRAC assays with the ethyl acetate extract, whereas *T. smyrnensis* flower extracts demonstrated higher antioxidant activity with the methanol extract in both assays.

The methanol extract of *T. gallica* flowers chelated iron (II) ions more effectively ($IC_{50} = 6$ mg/mL) compared to the leaf extracts ($IC_{50} = 8.3$ mg/mL). However, the chelating capacity of the organ extracts was not as high as that of the positive

**Table 1. Antioxidant activities of *Tamarix smyrnensis* flower extracts ($IC_{50} \pm$ standard deviation).**

|  | Methanol | Ethanol | Ethyl Acetate | P (Sig.) |
|---|---|---|---|---|
| DPPH (μg/mL) | $27.81 \pm 1.91$[b] | $25.15 \pm 2.07$[b] | $65.88 \pm 6.01$[a] | 0.0001 |
| Metal Chelating (mg/mL) | $49.84 \pm 2.83$[b] | $80.15 \pm 3.94$[a] | $174.12 \pm 4.55$[c] | 0.0001 |
| CUPRAC (μg/mL) | $35.95 \pm 2.23$[c] | $58.28 \pm 4.55$[b] | $151.46 \pm 3.56$[a] | 0.0005 |
| Standards | BHT $= 0.23 \pm 0.01$ | EDTA $= 5.30 \pm 0.44$ | Trolox $= 38.62 \pm 1.93$ |  |

Values are presented as mean $\pm$ standard deviation (SD). DPPH: 2,2-diphenyl-1-picrylhydrazyl; Metal Chelating: metal chelating activity; CUPRAC: cupric ion reducing antioxidant capacity; A significance level of $p < 0.05$ was applied, and one-way ANOVA followed by Tukey's test was performed. Different superscript letters within the same column indicate statistically significant differences.

control EDTA (IC$_{50}$ = 0.03 mg/mL) [26]. In our study, the methanol extract of *T. smyrnensis* flowers showed the highest iron-chelating activity, with an IC$_{50}$ value of 49.84 ± 2.83 mg/mL, whereas the organ extracts of *T. gallica* exhibited higher effectiveness. These findings suggest that *T. smyrnensis* extracts have the ability to bind iron ions. Moreover, the literature reports that plant extracts rich in phenolic compounds can form complexes with transition metal ions, stabilising them and preventing their participation in metal-catalysed initiation and hydroperoxide decomposition reactions [27].

**HPLC analysis result**

According to HPLC results of the *Tamarix* plant extract, ascorbic acid was detected only in the methanol extract at a concentration of 34.6 mg/L. The phenolic compound content of the extract was found to be highest in the methanol extract. Resveratrol was the only phenolic compound detected exclusively in the ethyl acetate extract, with a concentration of 264.9 mg/L. The flavonoid content of the extract was higher in the methanol and ethanol extracts, whereas rutin was detected only in the ethanol extract (131.7 mg/L) and myricetin only in the methanol extract (111.5 mg/L). In summary, the phenolic, ascorbic acid, and flavonoid contents of the extract vary depending on the solvent used (Table 2).

In our study, the *T. smyrnensis* methanol extract showed higher antioxidant activity compared to the other extracts. Therefore, methanol is the best solvent to extract flavonoids and phenolic compounds from medicinal plants [28]. It has been revealed that these phenolic and flavonoid compounds have strong antioxidant activity [29–31]. Studies have evaluated antioxidant activities and demonstrated significant differences between ethanol, methanol, and ethyl acetate [28,32]. The ethanol extract contains a higher concentration of chemical compounds compared to the ethyl acetate extract, and these findings also support the results of antioxidant activity in our study, indicating that the ethanol extract exhibited higher antioxidant activity compared to the ethyl acetate extract. This study is the first to evaluate

**Table 2. Quantitative distribution (mg/L) of vitamin, phenolic, and flavonoid compounds in methanol, ethanol, and ethyl acetate extracts.**

| No | Compounds | Methanol (mg/L) | Ethanol (mg/L) | Ethyl Acetate (mg/L) |
|---|---|---|---|---|
| Vitamins | | | | |
| 1 | Ascorbic acid | 34.60 | N/D | N/D |
| Phenolics | | | | |
| 2 | Gallic acid | 280.80 | N/D | 419.90 |
| 3 | 4-Hydroxybenzoic acid | 2253.90 | N/D | 67.30 |
| 4 | Vanillic acid | 581.80 | 263.10 | N/D |
| 5 | p-Coumaric acid | 2603.70 | 1395.90 | 740.80 |
| 6 | Ferulic acid | 495.50 | 309.50 | N/D |
| 7 | Rosmarinic acid | 126.80 | 69.90 | N/D |
| 8 | Pyrogallol | 1862.40 | 172.80 | 60.90 |
| 9 | Chlorogenic acid | 3087.60 | 427.40 | N/D |
| 10 | Resveratrol | N/D | N/D | 264.90 |
| Flavonoids | | | | |
| 11 | Catechin | 6974.40 | 94.10 | N/D |
| 12 | Epicatechin | 2981.80 | 358.60 | 117.30 |
| 13 | Rutin | N/D | 131.70 | N/D |
| 14 | Myricetin | 111.50 | N/D | N/D |
| 15 | Quercetin | 594.60 | 709.90 | 491.10 |
| 16 | Hesperetin | 163.40 | 321.40 | 48.80 |
| 17 | Baicalin | 64.20 | 70.50 | 57.40 |

*ND: Not detected

the antioxidant capacity of different extracts of *T. smyrnensis*. Flavonoids and total phenolic compounds are present in higher concentrations in the methanol extract compared to the other extracts; for example, 4-Hydroxybenzoic acid 2253.90 (mg/L), p-Coumaric acid 2603.70 (mg/L), Catechin 6974.40 (mg/L), Epicatechin 2981.80 (mg/L). The high phenolic and flavonoid contents indicate that these compounds may contribute to the antioxidant capacity of this plant, and when we compare the antioxidant activity and HPLC-identified phenolic, flavonoid, and vitamin results, the concentration levels of chemical compounds identified in the extracts obtained with different solvents are compatible with their antioxidant activities. The *T. smyrnensis* methanol extract has the potential to be a promising candidate as a natural plant source of high-value antioxidants.

*Tamarix* species are rich in phenolic acids, particularly in different plant parts, with woody tissues being especially abundant. Well-known phenolic acids such as ferulic acid, gallic acid, and p-coumaric acid are commonly found in *Tamarix* speciess [33]. For instance, ethyl acetate extracts obtained from the leaves and stems of *T. aphylla* were analyzed, and phenolic compounds including gallic, ferulic, and p-coumaric acids were identified for the first time in this species. The most abundant compounds in the leaves were quercetin (125.7±4.7 µg/g) and gallic acid (120.6±1.2 µg/g), whereas in the stems, gallic acid (24.3±3.3 µg/g) was the predominant compound [34]. HPLC analysis of *T. aphylla* leaf extract revealed the presence of phenolic and flavonoid compounds such as gallic acid, vanillic acid, p-coumaric acid, m-coumaric acid, ferulic acid, and quercetin [35].

The water and ethyl acetate extracts of *T. gallica* were analyzed using HPLC-UV/DAD, and phenolic acids such as quercetin and gallic acid were identified [23]. The chemical compounds identified in the flower extract of *T. smyrnensis* are consistent with our results, and varying amounts and types of chemical compounds have been detected in three different solvents (methanol, ethanol, and ethyl acetate). These findings indicate that solvent polarity significantly affects the extraction efficiency and compound profile, with methanol proving to be the most effective solvent for recovering both phenolic and flavonoid constituents in high quantities.

## Phenolic compound–protein interaction mapping based on STITCH predictions

The STITCH database was utilised to predict potential protein targets for selected phenolic compounds based on chemical similarity, literature mining, and known experimental associations. Among the compounds evaluated, 4-hydroxybenzoic acid, p-coumaric acid, chlorogenic acid, and catechin demonstrated the most relevant interactions with proteins involved in antioxidant defence and mitochondrial function (Table 3)

The compound 4-hydroxybenzoic acid showed a strong predicted interaction with COQ2 (score: 0.998), a key enzyme in the biosynthesis of coenzyme Q, an essential component of the mitochondrial respiratory chain. Similarly, it was also linked to PDSS1, PDSS2, and COQ6, all of which participate in the biosynthetic pathway of ubiquinone (CoQ10). These associations indicate the compound's potential involvement in mitochondrial antioxidant mechanisms. Catechin, a well-known flavanol, exhibited high-confidence interactions with several targets including HMOX1 (heme oxygenase 1, score: 0.865), PTGS2 (COX-2, score: 0.833), PON1 (score: 0.814), and DNMT1 (score: 0.837) (Fig 1).

These proteins are known to play roles in redox regulation, preventing lipid peroxidation, modulating inflammation, and influencing epigenetic processes. The interaction between catechin and HMOX1 is particularly significant due to the enzyme's role in heme degradation and cellular protection against oxidative damage. p-Coumaric acid showed moderate associations with overlapping targets, suggesting complementary activity with other phenolics in modulating oxidative stress pathways. Chlorogenic acid also appeared in the network, but with fewer strong direct interactions, potentially indicating a more supportive or synergistic role.

Overall, STITCH-based predictions provided a foundational dataset of compound–protein interactions, highlighting COQ2, HMOX1, PTGS2, and PON1 as central nodes related to antioxidants, likely modulated by plant-derived phenolics. These findings served as the basis for subsequent PPI (STRING) analysis and molecular docking simulations.

**Table 3. Antioxidant-Associated Protein Targets and STITCH Confidence Scores.**

| Protein Code | Gene Name | Function | References | STITCH Score |
|---|---|---|---|---|
| COQ2 | Coenzyme Q2 synthase | Involved in Coenzyme Q biosynthesis – key for mitochondrial antioxidant system | [36] | 0.998 |
| PDSS1/2 | Prenyltransferase subunits | Essential for ubiquinone (CoQ10) production | [37] | 0.990 |
| COQ6 | Coenzyme Q monooxygenase | Participates in the CoQ biosynthesis pathway | [38] | 0.989 |
| HMOX1 | Heme oxygenase 1 | Antioxidant defence; degradation of heme and scavenging of free radicals | [39] | 0.865 |
| DNMT1 | DNA methyltransferase 1 | Epigenetic regulation; potential indirect antioxidant relevance | [40,41] | 0.837 |
| PTGS2 (COX2) | Prostaglandin-endoperoxide synthase 2 | Key mediator of inflammation; involved in redox-sensitive signalling | [42] | 0.833 |
| PON1 | Prevents LDL oxidation | Prevents LDL oxidation; protects against oxidative damage | [43] | 0.814 |

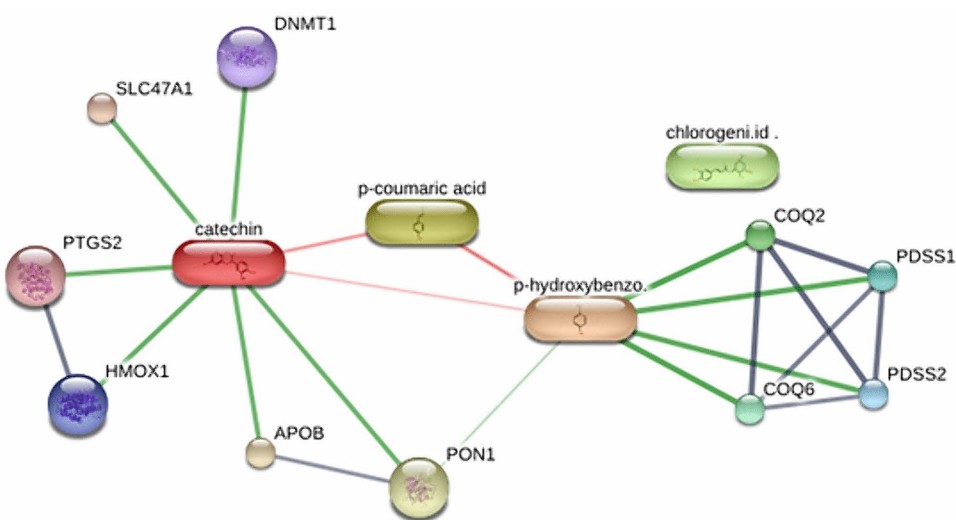

**Fig 1. Network view of selected phenolic compounds and their associated antioxidant-related protein targets based on STITCH.**

## Results of STRING-based protein–protein interaction analysis

Proteins identified via STITCH as potential targets of the selected phenolic compounds were further analysed using the STRING database to construct a protein–protein interaction (PPI) network (S2 Table). The analysis revealed a tightly connected cluster involving COQ2, COQ6, PDSS1, and PDSS2, all of which are central components of the mitochondrial coenzyme Q (CoQ10) biosynthesis pathway. These proteins showed high-confidence interactions (combined score > 0.98), suggesting coordinated activity in cellular antioxidant defence mechanisms. Additionally, HMOX1 was observed to interact with both PON1 and PTGS2 (COX-2), proteins known to be involved in oxidative stress response and inflammation. The inclusion of DNMT1, a protein linked to epigenetic regulation, also highlights the potential for indirect modulation of redox-sensitive pathways by phenolic compounds (Fig 2). Overall, the constructed network suggests that these compounds may influence multiple antioxidant and inflammation-related processes through both direct and indirect protein interactions.

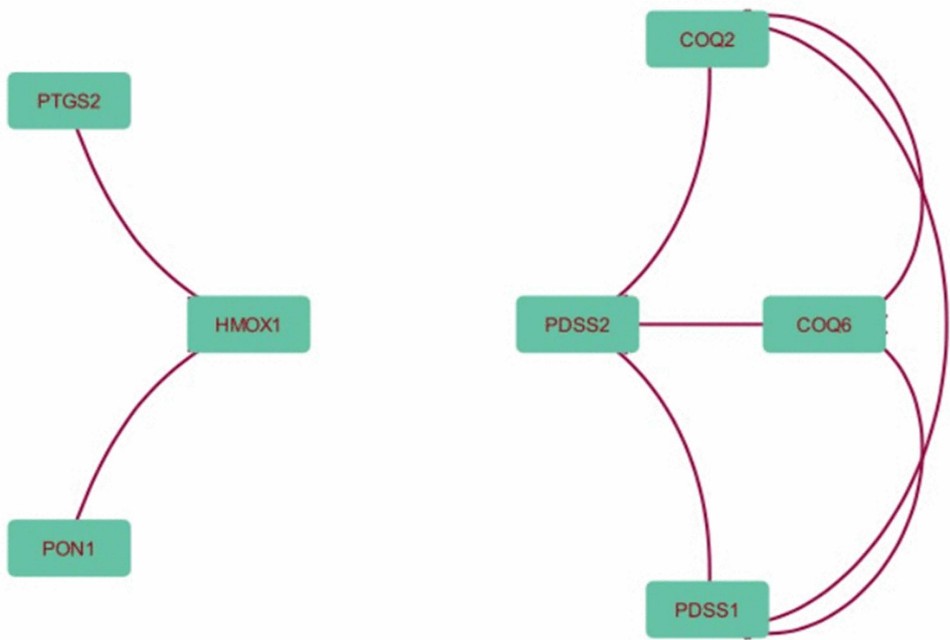

**Fig 2. Protein–protein interaction network of antioxidant-related targets constructed using STRING data.** Nodes represent proteins, and edges indicate high-confidence functional associations between them.

### Result of molecular docking

The binding modes and interactions of the compounds—hydroxybenzoic acid, p-coumaric acid, chlorogenic acid, and catechin—with the crystal structures of 1N45 (Human Heme Oxygenase-1, HO-1), 5IKQ (Human Cyclooxygenase-2, COX-2; PTGS2), and 1V04 (Serum Paraoxonase, PON1) were predicted using molecular docking, and the obtained scores are presented in Tables 4 and 5.

As observed, chlorogenic acid exhibited strong binding affinities with 1N45, 5IKQ, and 1V04, showing binding energies of −7.9, −9.3, and −8.5 kcal/mol, respectively. Similarly, catechin displayed high activity with binding energies of −8.0, −9.2, and −9.1 kcal/mol for 1N45, 5IKQ, and 1V04, respectively. In contrast, 4-hydroxybenzoic acid showed lower activity with binding energies of −5.6, −5.7, and −5.5 kcal/mol, while p-coumaric acid exhibited moderate activity with binding energies of −5.6, −6.6, and −5.9 kcal/mol (Table 4).

To confirm the reliability of the docking protocol, native and cross-redocking analyses were performed. The co-crystallized complex (5IKQ) reproduced its experimental pose with an RMSD = 0.380 Å. For apo structures, homolog-based validation was applied: 1N45 (homolog 3CZY) yielded RMSD = 1.290 Å, and 1V04 (homolog 3SRG, ligand 2-hydroxyquinoline) reproduced the crystallographic pose with RMSD = 0.800 Å (S3 Table). All RMSD values were below the 2 Å threshold, confirming the robustness and reproducibility of the docking procedure.

Analysis of protein–ligand interactions revealed notable details that explain the binding tendencies of the compounds (Table 5). Chlorogenic acid interacts with residues ARG44, CYS47, GLY45, CYS41, ASP126, PRO40, and VAL46 in chain B through multiple conventional hydrogen bonds (Fig 3). These hydrogen bonds involve both ligand–protein and protein–ligand interactions, contributing significantly to the stability of the complex. Furthermore, a π–cation interaction observed between ARG44 and the ligand provides additional electrostatic stabilization, enhancing the binding strength (S4 Table).

**Table 4. Molecular binding parameters of phenolic compounds against three target proteins.**

| | Protein | Binding Energy (kcal/mol) | LE | FQ | LLE | LELP | BEI | ΔG/ TPSA | Ki (µM) | pIC$_{50}$ |
|---|---|---|---|---|---|---|---|---|---|---|
| 4-hydroxybenzoic acid | 1N45 | −5.6 | 0.560 | 0.712 | 23.333 | −0.429 | 0.040 | 0.093 | 78.106 | 4.000 |
| | 51KQ | −5.7 | 0.570 | 0.724 | 23.750 | −0.421 | 0.041 | 0.094 | 65.969 | 4.071 |
| | 1V04 | −5.5 | 0.550 | 0.699 | 22.917 | −0.436 | 0.039 | 0.091 | 92.476 | 3.929 |
| p-coumaric acid | 1N45 | −5.6 | 0.467 | 0.643 | 35.000 | 0.342 | 0.034 | 0.093 | 78.106 | 4.000 |
| | 51KQ | −6.6 | 0.550 | 0.757 | 41.250 | 0.291 | 0.040 | 0.109 | 14.429 | 4.714 |
| | 1V04 | −5.9 | 0.492 | 0.679 | 36.875 | 0.325 | 0.035 | 0.098 | 47.060 | 4.214 |
| chlorogenic acid | 1N45 | −7.9 | 0.316 | 0.620 | 3.989 | 6.266 | 0.022 | 0.047 | 1.606 | 5.643 |
| | 51KQ | −9.3 | 0.372 | 0.729 | 4.697 | 5.319 | 0.026 | 0.055 | 0.151 | 6.643 |
| | 1V04 | −8.5 | 0.340 | 0.668 | 4.293 | 5.824 | 0.024 | 0.051 | 0.583 | 6.071 |
| catechin | 1N45 | −8.0 | 0.381 | 0.703 | 5.161 | 4.073 | 0.027 | 0.072 | 1.356 | 5.714 |
| | 51KQ | −9.2 | 0.438 | 0.808 | 5.935 | 3.537 | 0.031 | 0.083 | 0.179 | 6.643 |
| | 1V04 | −9.1 | 0.433 | 0.799 | 5.871 | 3.577 | 0.031 | 0.082 | 0.212 | 6.571 |

*BEI: Binding Efficiency Index, ΔG/ TPSA: Binding Energy per TPSA, FQ: Fit Quality, Ki: Estimated Inhibition Constant, LE: Ligand Efficiency, LELP: Ligand Efficiency–Lipophilicity, LLE: Lipophilic Ligand Efficiency, pIC$_{50}$: Negative Logarithm of IC$_{50}$.

**Table 5. Key molecular interactions between ligands and PTGS2 (5IKQ).**

| Compounds | Protein | H-Bond | Pi–Pi Stacking | Alkyl Interactions |
|---|---|---|---|---|
| Chlorogenic acid | 51KQ | B:ARG44:HN – [001:O7], B:CYS47:HN – [001:O2], [001:H1] – B:GLY45:O, [001:H2] – B:CYS41:O, [001:H3] – B:CYS41:O, [001:H13] – B:ASP126:OD2, B:PRO40:HD1 – [001:O3], B:VAL46:HA – [001:O5], [001:H11] – B:CYS41:O, B:ARG44:HE – [001] (Pi-Donor H-Bond) | – | B:ARG44:NH2 – [001] (Pi-Cation) |
| Catechin | 51KQ | A:ARG469:HE – [001:O5], [001:H2] – A:LYS468:O, [001:H3] – A:CYS41:O, [001:H9] – A:ASP125:OD2, [001:H10] – A:ASP125:OD2, A:ARG44:HA – [001:O1], A:ARG44:HD1 – [001:O1], [001:H8] – A:GLY45:O, A:ARG44:HN – [001] (Pi-Donor H-Bond) | – | A:LEU152 (Alkyl), [001] – A:LYS468 (Pi-Alkyl), [001] – A:ARG44 (Pi-Alkyl) |

Catechin, on the other hand, forms hydrogen bonds with ARG469, LYS468, CYS41, ASP125, GLY45, and ARG44 in chain A (Fig 4). In addition, catechin exhibits hydrophobic interactions within the binding site, including alkyl contact with LEU152 and π–alkyl interactions between LYS468 and ARG44. This interaction profile demonstrates the high affinity of catechin for the binding site, with both polar and apolar contributions playing important roles in stabilizing the complex (S5 Table).

Chlorogenic acid and catechin compounds exhibited strong antioxidant activities, which may be attributed to their phenolic and flavonoid structures. Molecular docking studies revealed that these compounds have high binding affinity for the COX-2 active site (chlorogenic acid: 5IKQ, COX-2; −9.3 kcal/mol; catechin: 5IKQ, −9.2 kcal/mol). The obtained results support the scientific rationale for using *T. smyrnensis* in the treatment of disorders such as asthma, rheumatoid arthritis, cancer, and diabetes, which are activated by oxidative stress-induced inflammation. In this study, the compound 6α-hydroxy-4[14],10[15]-guainadien-8α,12-olide also showed good binding affinity with COX-2 (docking score: −8.98; Glide binding energy: −36.488 kcal/mol), shedding light on its potential anti-inflammatory mechanism [44]. Another study reported that spirofuran-triazolo[1,5-a] pyrimidine derivatives (5a–i) could effectively bind to the COX-2 enzyme according to modeling results (5′R,6′S,7′R; −6.48 to −7.67 kcal/mol and 5′S,6′S,7′S; −6.53 to −8.42 kcal/mol) [40]. Our study further suggests that natural antioxidants, such as the flower extract of *T. smyrnensis*, can be considered as novel therapeutic options for the treatment of various inflammatory disorders.

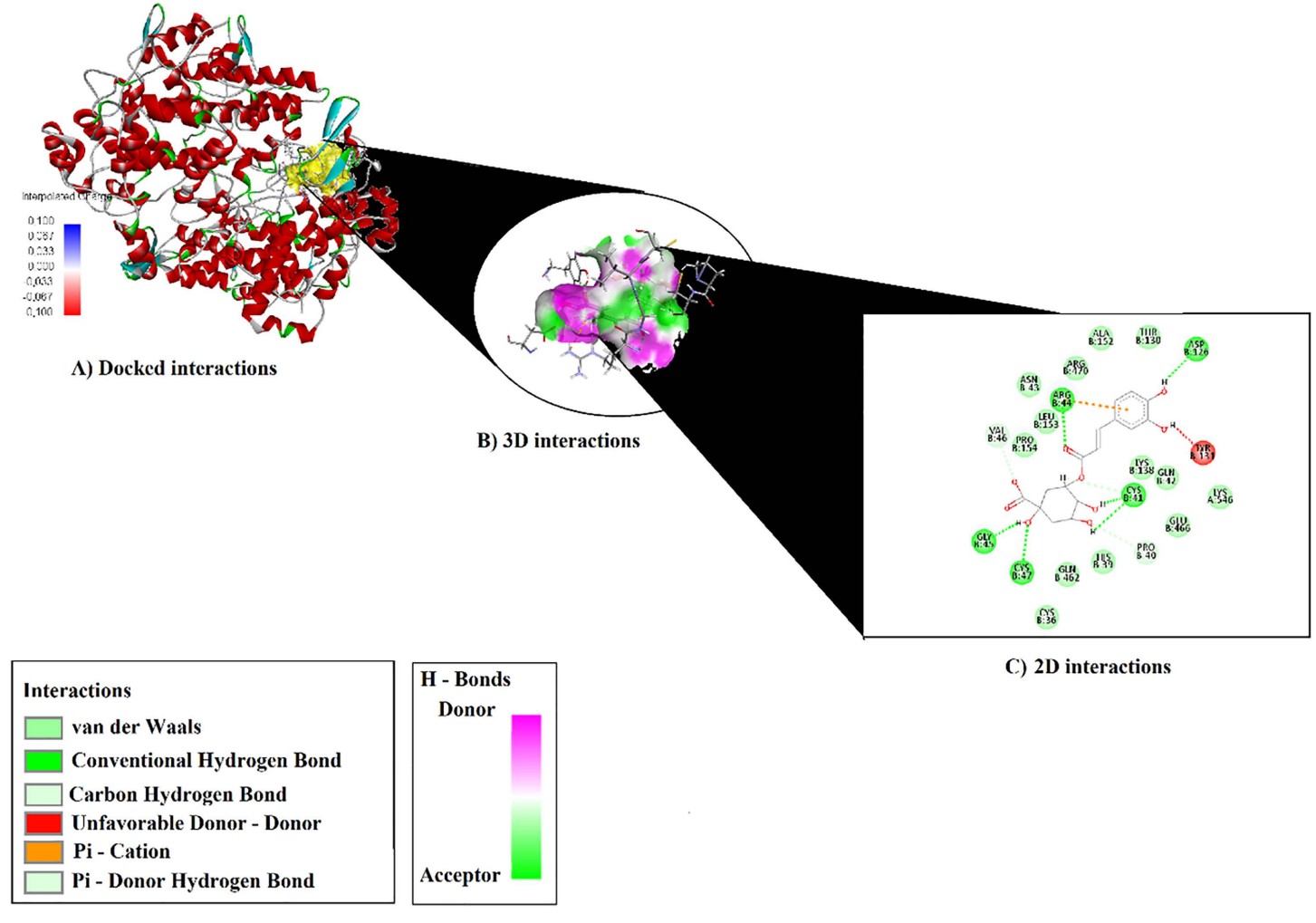

**Fig 3. Binding Representations of Chlorogenic Acid with 51KQ: Docked (A), 3D (B), and 2D (C) Views.**

Inducible heme oxygenase (HO-1) is a critical antioxidant stress protein primarily induced by reactive oxygen species (ROS), cytokines, and hyperthermia [45]. Modulation of HO-1 expression and activity represents an important mechanism for maintaining cellular redox homeostasis and preventing oxidative stress-induced damage. In a previous study, binding energies for quercetin and rutin were reported as −6.18 kcal/mol and −5.26 kcal/mol, respectively [46], whereas in the present study, these values were determined as −7.9 kcal/mol and −8.0 kcal/mol for chlorogenic acid and catechin, respectively.

These findings indicate that chlorogenic acid and catechin exhibit lower binding energies with HO-1, suggesting stronger interactions compared to quercetin and rutin. Considering the antioxidant and cytoprotective functions of HO-1, such strong interactions imply that chlorogenic acid and catechin may enhance the capacity to protect cells against oxidative stress. Furthermore, the potential of these compounds to stimulate HO-1 expression and activity positions them as significant pharmacological and therapeutic targets. In this context, the role of flavonoid and phenolic compounds in HO-1-mediated antioxidant mechanisms may provide valuable insights for the development of novel strategies aimed at preventing or treating oxidative stress-related disorders.

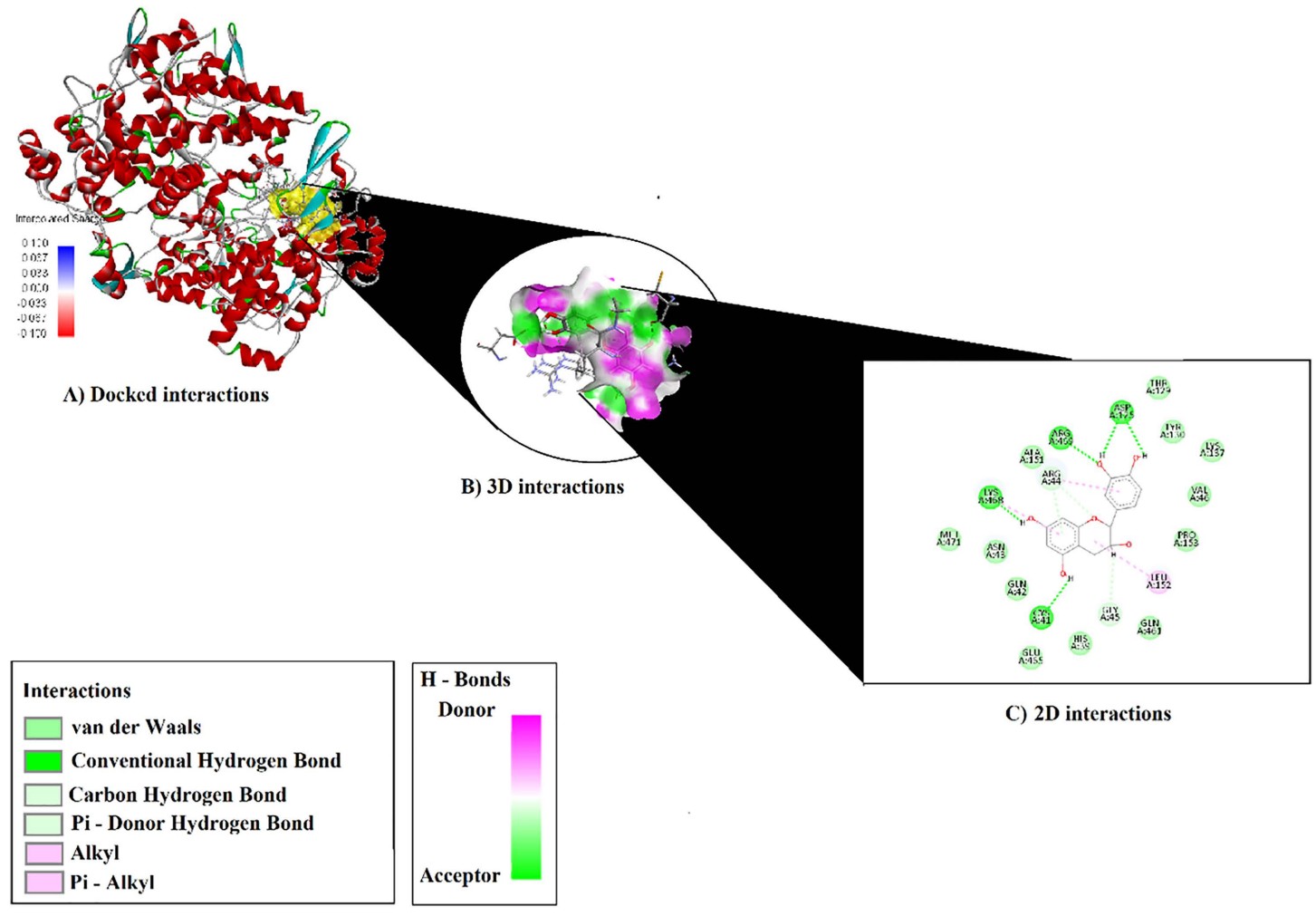

**Fig 4. Binding Representations of catechin with 51KQ: Docked (A), 3D (B), and 2D (C) Views.**

Interest in the role of Paraoxonase 1 (PON1) in drug metabolism has increased in recent years [47]. Numerous studies have reported interactions between PON1 and various drugs or biologically active compounds. These studies primarily focus on the pharmacological aspects of PON1 and its interactions with different drugs, as PON1 has been associated with various diseases. Research on the modulation of human serum PON1 activity by drugs has shown that compounds such as analgesics [48], anesthetics [49], calcium channel blockers [50], chemotherapeutics [51], and antiepileptics [52] can reduce PON1 activity. Similarly, sulfonamides [53] and dihydroxycoumarin [54] derivatives have been reported to exert inhibitory effects on human PON1 (hPON1). In the present study, molecular docking analyses demonstrated that chlorogenic acid and catechin exhibit strong binding affinity toward PON1, with binding energies of −8.5 kcal/mol and −9.1 kcal/mol, respectively. These findings suggest that chlorogenic acid and catechin may act as potential hPON1 inhibitors.

## Conclusions

The results of the metal chelating and CUPRAC antioxidant analyses demonstrated that ethanol-derived *T. smyrnensis* extracts exhibited the highest antioxidant activity. Additionally, the extracts prepared with ethanol and methanol showed

very similar activity in the DPPH antioxidant assay performed in our study. In contrast, extracts obtained using ethyl acetate demonstrated lower antioxidant activity compared to the other solvents. Depending on the solvent used, the antioxidant activities of *T. smyrnensis* extracts varied, indicating that solvent selection plays a significant role in determining their biological efficacy. In the methanol extract of *T. smyrnensis*, high concentrations of phenolic and flavonoid compounds such as 4-hydroxybenzoic acid, p-coumaric acid, and catechin were determined. The interactions of these compounds with antioxidant-related proteins were predicted using the STITCH database, and it was found that key proteins involved in mitochondrial function and oxidative stress response, including COQ2, HMOX1, PTGS2, and PON1, were affected by these compounds. Molecular docking studies further revealed that chlorogenic acid and catechin exhibited strong binding affinities toward PTGS2, indicating their potential therapeutic value in targeting oxidative stress pathways. These findings only predict potential effects and need validation through in vivo and in vitro experiments. Future studies should confirm the biological activities of these compounds, assess their toxicological profiles, and explore their specific pharmacological applications, taking into account the study's limitations.

## Supporting information

**S1 Table. Antioxidant activity results of plant extracts (n = 3).**
(DOCX)

**S2 Table. STRING network interaction data.**
(CSV)

**S3 Table. Docking validation summary showing RMSD values of ligands re-docked into their respective active sites, confirming protocol reliability.**
(DOCX)

**S4 Table. Interaction summary of chlorogenic acid with the 51KQ active site.**
(XLSX)

**S5 Table. Interaction summary of catechin with the 51KQ active site.**
(XLSX)

## Author contributions

**Conceptualization:** Erdi Can Aytar, Alper Durmaz.

**Data curation:** Mika Sillanpää.

**Formal analysis:** Erdi Can Aytar.

**Investigation:** Erdi Can Aytar, Saleh Al-Farraj, Alper Durmaz, Mika Sillanpää.

**Methodology:** Erdi Can Aytar, Abidin Gümrükçüoğlu, Saleh Al-Farraj.

**Resources:** Emine İncilay Torunoğlu, Abidin Gümrükçüoğlu, Saleh Al-Farraj, Alper Durmaz, Mika Sillanpää.

**Software:** Abidin Gümrükçüoğlu.

**Supervision:** Erdi Can Aytar.

**Validation:** Erdi Can Aytar, Emine İncilay Torunoğlu.

**Visualization:** Erdi Can Aytar, Emine İncilay Torunoğlu.

**Writing – original draft:** Erdi Can Aytar, Emine İncilay Torunoğlu, Mika Sillanpää.

**Writing – review & editing:** Erdi Can Aytar, Emine İncilay Torunoğlu, Mika Sillanpää.

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
