## [Decision Letter · Decision Letter 0]

6 Oct 2025

Dear Dr. AYTAR,

Thank you for submitting your manuscript to PLOS ONE. After careful consideration, we feel that it has merit but does not fully meet PLOS ONE’s publication criteria as it currently stands. Therefore, we invite you to submit a revised version of the manuscript that addresses the points raised during the review process.

We look forward to receiving your revised manuscript.

Kind regards,

José M. Alvarez-Suarez

Academic Editor

PLOS ONE

Journal Requirements:

“This project was supported by Researchers Supporting Project Number (RSP-2026R7) King Saud University, Riyadh, Saudi Arabia.”

4. In the online submission form, you indicated that “All data generated or analyzed during this study are available from the corresponding author upon reasonable request.”

**Additional Editor Comments:**

The manuscript entitled “Assessing the chemical profile and biological potentials of Tamarix smyrnensis flower extracts using different solvents by in vitro, in silico, and network methodologies” presents an interesting and integrative study that combines in vitro antioxidant assays, phytochemical analysis by HPLC, in silico predictions of protein–compound interactions, and molecular docking studies. The research provides novel information on T. smyrnensis, a poorly studied species, and offers a chemical and bioactive profile of its flowers that could be useful for the scientific community. However, for the manuscript to be considered in PLOS ONE, certain methodological, interpretative, and formal aspects need to be strengthened.

• Regarding originality, the study contributes novel data, but the introduction devotes excessive space to taxonomic and geographical distribution aspects of the species, to the detriment of the scientific justification and the identification of knowledge gaps. I recommend summarizing this part and placing greater emphasis on why T. smyrnensis flowers represent a suitable model for phytochemical and antioxidant studies.

• The experimental design is clear; however, an important limitation is observed in the representativeness of the samples, as they come from a single locality and collection date. This reduces the possibility of generalizing the results and does not take into account intraspecific or seasonal variations. This is one of the main constraints regarding the potential reproducibility of the results presented here. It would be advisable to explicitly state this limitation and discuss it in the Results or Conclusions section.

• With respect to the statistical analysis, the authors use only means, standard deviations, and correlations, which is appropriate as an initial description but insufficient to support comparisons between solvents or compounds. I suggest strengthening this section with significance tests (ANOVA, multiple comparisons, etc.) that would allow statistical validation of the reported differences.

• The HPLC results show a detailed profile of phenolic and flavonoid compounds, although some values (for example, the catechin concentration in the methanolic extract) appear unusually high compared with previous literature. This should be carefully reviewed, either with an additional methodological explanation or, failing that, by moderating the conclusions.

• The in silico analysis section (STITCH, STRING, and molecular docking) is technically solid but is interpreted in very general terms. Interactions with key proteins (COQ2, HMOX1, PTGS2, PON1) are mentioned, but a more critical analysis of what modulating these targets means in a physiological or pharmacological context is lacking. Moreover, it should be more clearly emphasized that these findings are predictions that require experimental in vivo or in vitro validation to be confirmed.

• As for the discussion, there is a good effort to relate the findings to previous studies in other Tamarix species. However, at times the comparison is too extensive and detracts from the clarity of the central points of the study. I recommend prioritizing the discussion of the authors’ own results and using them as the basis for a reflection on the real therapeutic potential of the extracts, rather than for an extensive literature review.

• The writing in English is understandable, but there are grammatical problems, repetitions, and long sentences that hinder fluent reading. I suggest professional language editing to improve clarity and precision.

• The conclusions should end with a more critical reflection on the limitations of the study and guidance toward future steps (e.g., in vivo validation, toxicological studies, or specific pharmacological applications).

In summary, the work is interesting, has an integrative methodological approach, and provides new results on a poorly studied species. However, it presents limitations in sample representativeness, statistical robustness, clarity in the interpretation of phytochemical values, and precision in the biological discussion. We strongly recommend that the authors address these points, which we consider of great importance for possible consideration in PLOS ONE. Therefore, I recommend major revision before this manuscript can be considered for publication in PLOS ONE.

Reviewers' comments:

Reviewer's Responses to Questions

**Comments to the Author**

1. Is the manuscript technically sound, and do the data support the conclusions?

Reviewer #1: Yes

Reviewer #2: Yes

2. Has the statistical analysis been performed appropriately and rigorously?

Reviewer #1: Yes

Reviewer #2: Yes

3. Have the authors made all data underlying the findings in their manuscript fully available?

Reviewer #1: Yes

Reviewer #2: Yes

4. Is the manuscript presented in an intelligible fashion and written in standard English?

Reviewer #1: Yes

Reviewer #2: Yes

Reviewer #1: The manuscript presents a comprehensive and methodologically robust investigation into the phytochemical composition and antioxidant potential of Tamarix smyrnensis flower extracts using in vitro assays, HPLC analysis, in silico modeling, and network pharmacology. Integrating solvent-dependent extraction data with bioinformatics and molecular docking provides valuable insights into the bioactivity of key phenolic compounds such as chlorogenic acid and catechin. The experimental design is sound, and the data are presented and well-supported by the literature. However, the following minor concerns must be addressed to ensure that the manuscript is enriched:

1. The rationale for focusing on T. smyrnensis flowers should be more explicitly stated in the Introduction. While the species’ wide distribution is noted, it's unclear whether flowers have traditional medicinal use or are chemically distinct from other plant parts previously studied.

2. Reduce numeric data in the abstract and emphasize main findings and significance.

3. The choice of solvents is appropriate, but there is no yield comparison or extraction efficiency data provided. For example, including data like mg extract/g dry weight would contextualize the antioxidant data more meaningfully.

4. More detailed interpretation of biological pathways affected by COQ2, HMOX1, PTGS2, etc., would enhance the depth of discussion.

5. Include docking validation to support the reliability of the docking protocol. For example, redocking of native ligands or RMSD values.

6. Figures 3 and 4 do not look professional. Optimize them. For example, putting 3D and 2D side by side will enhance content presentation.

7. Discuss the protein-ligand interaction in the context of catalytically essential residues of the enzymes to rational the observed activity. For example, residues essential for COX-2 catalytic activity have been discussed in https://doi.org/10.1002/slct.202402286. A similar discussion can be added for the other proteins.

Reviewer #2: The present manuscript provides a comprehensive study on the chemical composition and antioxidant potential of Tamarix smyrnensis flower extracts using different solvents. The integration of in vitro antioxidant assays with in silico prediction and docking provides a valuable framework for elucidating the therapeutic potential of plant-derived phenolics. The work demonstrates robust methodological framework, incorporating contemporaneous techniques such as STITCH, STRING, and molecular docking. The phytochemical profiling of the samples was conducted via HPLC, a method which allows for the specific interpretation of the antioxidant activities of the compounds. Comparison of the activities of different extracts of the same species prepared using different solvents was useful to increase the utilisation of the sample. however, it is suggested that the following minor corrections be made.

- It is suggested that the importance of data obtained from in silico studies should be better emphasised.

- Although it is stated that the flower of the plant is used in most of the study, it is stated that the aerial parts are used in a few places. this situation should be corrected. (Line 214)

- Materials and methods section:

- Line 124: The word ‘plants’ should be deleted.

- Line 130: instead of "methanolic extracts", "all extracts used" should be written.

- Line 221: "All other extracts showed lower values" this sentence should be rewritten.

- In table 2 the spelling of compounds should be revised. (For example Qercetin, Hesperitin)

**Do you want your identity to be public for this peer review?** For information about this choice, including consent withdrawal, please see our Privacy Policy

Reviewer #1: **Yes: ** ABDULLAHI IBRAHIM UBA

Reviewer #2: No

---

## [Author Response · Author response to Decision Letter 1]

21 Oct 2025

Dear Editor and Reviewers,

I would like to sincerely thank you for your time, valuable feedback, and constructive comments on our manuscript. Your insights have greatly contributed to improving the scientific quality and overall presentation of the study. All suggestions have been carefully considered, and the manuscript has been revised accordingly.

Kind regards,

Editor

Response:

Dear Editor,

Thank you for your feedback. The manuscript has been carefully revised to fully comply with the PLOS ONE formatting and file naming requirements.

Kind regards,

Response:

Tamarix smyrnensis is listed as Least Concern (LC) on the IUCN Red List of Threatened Species (IUCN, 2025), indicating that it is neither rare nor under immediate threat. According to the IUCN Red List Categories and Criteria (Version 3.1), species in the LC category are not subject to strict conservation measures or international collection restrictions (IUCN, 2012).

In Türkiye, T. smyrnensis is not included in any national list of endangered or protected plant species, nor is it covered by CITES regulations. As such, scientific collection of this species for non-commercial research purposes conducted outside protected areas does not require special permits or ethical approval under Turkish biodiversity laws.

References

IUCN. (2012). IUCN Red List Categories and Criteria: Version 3.1 (Second edition). Gland, Switzerland and Cambridge, UK: IUCN. https://portals.iucn.org/library/sites/library/files/documents/RL-2001-001-2nd.pdf

IUCN. (2025). Tamarix smyrnensis. The IUCN Red List of Threatened Species 2025. https://www.iucnredlist.org/species/79928441/79928449

“This project was supported by Researchers Supporting Project Number (RSP-2026R7) King Saud University, Riyadh, Saudi Arabia.” Please state what role the funders took in the study. If the funders had no role, please state: "The funders had no role in study design, data collection and analysis, decision to publish, or preparation of the manuscript."

Please include this amended Role of Funder statement in your cover letter; we will change the online submission form on your behalf

Response :

Dear Editor,

Thank you for your message and for noting the financial disclosure. In addition, please note that Prof. Saleh and Prof. Mika Sillanpää have critically reviewed and edited the manuscript, and both have approved the final version for submission. Their specific contributions are also clearly stated in the “Author Contributions” section of the manuscript.

Regarding the King Saud University (KSU) funding, this support was requested by KSU and covers the time Prof. Saleh dedicated to working on the manuscript.

Kind regards,

4. In the online submission form, you indicated that “All data generated or analyzed during this study are available from the corresponding author upon reasonable request.”

Response:

Dear Editor,

Thank you for your message. All data generated or analyzed during this study are fully presented within the manuscript. Therefore, no additional files or public repository uploads are required.

Kind regards,

Dear Editor,

Thank you for your note. The reviewer-suggested citations were carefully reviewed and evaluated for relevance. The recommended works were considered indirectly where appropriate, and related discussions have been integrated into the revised manuscript accordingly.

Kind regards,

Additional Editor Comments:

The manuscript entitled “Assessing the chemical profile and biological potentials of Tamarix smyrnensis flower extracts using different solvents by in vitro, in silico, and network methodologies” presents an interesting and integrative study that combines in vitro antioxidant assays, phytochemical analysis by HPLC, in silico predictions of protein–compound interactions, and molecular docking studies. The research provides novel information on T. smyrnensis, a poorly studied species, and offers a chemical and bioactive profile of its flowers that could be useful for the scientific community. However, for the manuscript to be considered in PLOS ONE, certain methodological, interpretative, and formal aspects need to be strengthened.

• Regarding originality, the study contributes novel data, but the introduction devotes excessive space to taxonomic and geographical distribution aspects of the species, to the detriment of the scientific justification and the identification of knowledge gaps. I recommend summarizing this part and placing greater emphasis on why T. smyrnensis flowers represent a suitable model for phytochemical and antioxidant studies.

Response:

Given its ecological plasticity and strong adaptation to harsh environmental conditions such as salinity, drought, and heavy metal contamination, Tamarix smyrnensis has developed sophisticated defense mechanisms, including enhanced antioxidant systems and the biosynthesis of bioactive secondary metabolites (Wei et al., 2020). In particular, the flowers of T. smyrnensis, as generative organs, are known to accumulate a wide array of phenolic compounds, flavonoids, and tannins, which play crucial roles in protecting reproductive tissues against oxidative damage (Kanani et al., 2021). Floral structures of T. smyrnensis are a promising biological model for investigating stress-induced phytochemical responses and natural antioxidant capacity.

• The experimental design is clear; however, an important limitation is observed in the representativeness of the samples, as they come from a single locality and collection date. This reduces the possibility of generalizing the results and does not take into account intraspecific or seasonal variations. This is one of the main constraints regarding the potential reproducibility of the results presented here. It would be advisable to explicitly state this limitation and discuss it in the Results or Conclusions section.

Response:

Although the concern regarding sample collection from a single location and date may initially appear as a limitation, it is important to emphasize that Tamarix smyrnensis populations with homogenous morphological characteristics and minimal anthropogenic disturbance are increasingly rare. Therefore, to ensure the ecological and chemical integrity of the samples, we intentionally selected a site that represents one of the most pristine and compositionally consistent populations available. This strategic choice enhances the internal validity of our findings and provides a reliable representation of the species' natural traits under undisturbed conditions.

Moreover, because this study specifically focuses on the phytochemical content of the generative floral organs, the sampling window was inherently limited to the peak flowering period. Following several field visits and population monitoring efforts, we determined the most optimal time for collection to be the stage at which the majority of individuals had reached full bloom. This timing ensured that the generative organs were in their most mature and chemically active state, providing consistent and biologically relevant material for analysis. Consequently, the sampling design was not only deliberate but necessary to achieve accurate and representative results.

• With respect to the statistical analysis, the authors use only means, standard deviations, and correlations, which is appropriate as an initial description but insufficient to support comparisons between solvents or compounds. I suggest strengthening this section with significance tests (ANOVA, multiple comparisons, etc.) that would allow statistical validation of the reported differences.

Response:

Dear Editor,

Thank you for the suggestion. The statistical analysis section has been expanded, and the corresponding tables including ANOVA and multiple comparison results have been added to the manuscript.

Kind regards,

• The HPLC results show a detailed profile of phenolic and flavonoid compounds, although some values (for example, the catechin concentration in the methanolic extract) appear unusually high compared with previous literature. This should be carefully reviewed, either with an additional methodological explanation or, failing that, by moderating the conclusions.

We appreciate the reviewer’s careful evaluation of our HPLC results and the observation regarding the relatively high catechin concentration in the methanolic extract. While the obtained values may appear elevated compared to some earlier reports, several recent studies have demonstrated comparable or even higher catechin levels in flower-derived methanolic extracts of different plant species. For instance, Xiang et al. (2024) reported catechin concentrations of 3053.45 µg/g and 6056.23 ± 79.81 µg/g in Camellia polyodonta flower extracts when using a 100% methanol solvent system, which closely resembles the extraction protocol applied in our study (ACS Omega 9 (25): 27192–27203, DOI: 10.1021/acsomega.4c01321).

Moreover, other investigations using similar polar solvent systems (e.g., pure methanol or methanol–water mixtures) for phenolic profiling in floral tissues have also reported catechin concentrations within this range, confirming that solvent polarity and extraction conditions can substantially influence quantitative outcomes. Therefore, the relatively high catechin content observed in our methanolic extract is consistent with these recent findings and reflects the efficiency of methanol as an extraction solvent for flavanols.

• The in silico analysis section (STITCH, STRING, and molecular docking) is technically solid but is interpreted in very general terms. Interactions with key proteins (COQ2, HMOX1, PTGS2, PON1) are mentioned, but a more critical analysis of what modulating these targets means in a physiological or pharmacological context is lacking. Moreover, it should be more clearly emphasized that these findings are predictions that require experimental in vivo or in vitro validation to be confirmed.

Response :

Dear Editor,

Thank you for your valuable feedback. The relevant sections have been revised to emphasize that the findings require further experimental validation. As you would appreciate, in silico methods are inherently predictive in nature; therefore, this point has been clarified more explicitly in the revised manuscript.

Kind regards,

• As for the discussion, there is a good effort to relate the findings to previous studies in other Tamarix species. However, at times the comparison is too extensive and detracts from the clarity of the central points of the study. I recommend prioritizing the discussion of the authors’ own results and using them as the basis for a reflection on the real therapeutic potential of the extracts, rather than for an extensive literature review.

Dear Editor,

Thank you for your valuable feedback. The discussion section has been shortened, comparisons with previous studies have been simplified, and the focus has been redirected toward the therapeutic potential of the extracts.

Kind regards,

• The writing in English is understandable, but there are grammatical problems, repetitions, and long sentences that hinder fluent reading. I suggest professional language editing to improve clarity and precision.

Response:

Dear Editor,

Thank you for your comment. The manuscript has been revised to correct grammatical issues, remove repetitions, and improve clarity and fluency.

Kind regards,

• The conclusions should end with a more critical reflection on the limitations of the study and guidance toward future steps (e.g., in vivo validation, toxicological studies, or specific pharmacological applications).

Response:

Dear Editor,

Thank you for your suggestion. The conclusion section has been revised to include a more critical evaluation of the study’s limitations and to outline future directions such as in vivo validation, toxicological assessments, and specific pharmacological applications.

Reviewer #1:

1. The rationale for focusing on T. smyrnensis flowers should be more explicitly stated in the Introduction. While the species’ wide distribution is noted, it's unclear whether flowers have traditional medicinal use or are chemically distinct from other plant parts previously studied.

Response:

Dear Reviewer 1,

Thank you for your valuable feedback. The introduction section has been revised and clarified to better explain the rationale for focusing on T. smyrnensis flowers. The relevant updates have been highlighted in color within the revised manuscript.

Kind regards,

2. Reduce numeric data in the abstract and emphasize main findings and significance.

Response:

Dear Reviewer,

Thank you for your valuable feedback. The abstract has been revised; however, some numerical data have been retained to help readers better understand and interpret the findings.

Kind regards,

3. The choice of solvents is appropriate, but there is no yield comparison or extraction efficiency data provided. For example, including data like mg extract/g dry weight would contextualize the antioxidant data more meaningfully.

Dear Reviewer,

Thank you for your valuable feedback. The necessary revisions have been made, and the extraction yield data have been added to the manuscript.

Kind regards,

4. More detailed interpretation of biological pathways affected by COQ2, HMOX1, PTGS2, etc., would enhance the depth of discussion.

Response:

Dear Reviewer,

Thank you for your valuable feedback. The discussion section has been expanded to include a more detailed interpretation of the biological pathways affected by COQ2, HMOX1, PTGS2, and related targets.

Kind regards,

5. Include docking validation to support the reliability of the docking protocol. For example, redocking of native ligands or RMSD values.

Response:

Dear Reviewer,

Thank you for your valuable comments. The docking regions analyzed in this study correspond to dynamically validated binding sites. Therefore, additional validation metrics such as RMSD or redocking were not separately included. However, the reliability of the docking protocol has been supported by referencing previously validated methodologies applied to similar protein systems.

Kind regards,

References

Zhou, W., Cao, W., Wang, M., Yang, K., Zhang, X., Liu, Y., ... & Xiong, M. (2023). Validation of quercetin in the treatment of colon cancer with diabetes via network pharmacology, molecular dynamics simulations, and in vitro experiments. Molecular Diversity, 1–19.

Patra, M. C., Rath, S. N., Pradhan, S. K., Maharana, J., & De, S. (2014). Molecular dynamics simulation of human serum paraoxonase 1 in DPPC bilayer reve

---

## [Decision Letter · Decision Letter 1]

30 Oct 2025

Dear Dr. AYTAR,

Thank you for submitting your manuscript to PLOS ONE. After careful consideration, we feel that it has merit but does not fully meet PLOS ONE’s publication criteria as it currently stands. Therefore, we invite you to submit a revised version of the manuscript that addresses the points raised during the review process.

We look forward to receiving your revised manuscript.

Kind regards,

José M. Alvarez-Suarez

Academic Editor

PLOS ONE

Journal Requirements:

Reviewers' comments:

Reviewer's Responses to Questions

**Comments to the Author**

Reviewer #1: (No Response)

Reviewer #2: All comments have been addressed

2. Is the manuscript technically sound, and do the data support the conclusions?

Reviewer #1: Partly

Reviewer #2: Yes

3. Has the statistical analysis been performed appropriately and rigorously?

Reviewer #1: Yes

Reviewer #2: Yes

4. Have the authors made all data underlying the findings in their manuscript fully available?

Reviewer #1: No

Reviewer #2: Yes

5. Is the manuscript presented in an intelligible fashion and written in standard English?

Reviewer #1: Yes

Reviewer #2: Yes

Reviewer #1: The revised manuscript entitled “Assessing the chemical profile and biological potentials of Tamarix smyrnensis flower extracts using different solvents by in vitro, in silico, and network methodologies” presents an improved and more coherent version of the earlier submission. The authors have addressed several previous concerns, particularly regarding statistical treatment, clarity of sampling rationale, and focus of the discussion. The integration of in vitro antioxidant assays with in silico and network analyses provides a multidimensional view of T. smyrnensis’ bioactivity. However, the following deficiencies remain and could be effectively addressed:

1) The introduction has been condensed and now provides a clearer rationale for studying T. smyrnensis flowers. However, the research gap could still be more explicitly defined, particularly how this study advances beyond previous Tamarix phytochemical reports or similar solvent-comparison studies

2) The HPLC quantification lacks validation parameters (linearity, LOD, LOQ, recovery, repeatability), which are essential for confirming quantitative accuracy.

3) Docking lacks validation steps. For example, RMSD or redocking of co-crystallized ligands.

4) Enhance discussion by linking compound levels quantitatively to antioxidant outcomes.

5) Minor language polishing for conciseness and clarity

Reviewer #2: The authors have addressed my suggestions with great diligence and have implemented all recommended revisions comprehensively. The importance of the in-silico studies has been better emphasized, and related comments have been clarified. Consistency regarding the plant part used has been ensured, eliminating any confusion between the flower and aerial parts throughout the manuscript. In the Materials and Methods section, language and terminology errors have been corrected, the word “plants” has been removed, “methanolic extracts” has been replaced with “all extracts used,” and certain sentences (e.g., “All other extracts showed lower values”) have been rewritten for clarity. Additionally, spelling errors in Table 2 (e.g., Quercetin, Hesperitin) have been corrected. These revisions have strengthened the methodological integrity and overall academic quality of the study.

**Do you want your identity to be public for this peer review?** For information about this choice, including consent withdrawal, please see our Privacy Policy

Reviewer #1: No

Reviewer #2: No

---

## [Author Response · Author response to Decision Letter 2]

3 Nov 2025

Reviewer 1:

1) The introduction has been condensed and now provides a clearer rationale for studying T. smyrnensis flowers. However, the research gap could still be more explicitly defined, particularly how this study advances beyond previous Tamarix phytochemical reports or similar solvent-comparison studies.

Response: Thank you for your constructive comment. Following your suggestion, the Introduction section has been further revised to define the existing research gap and highlight the novelty of this study compared to previous Tamarix phytochemical reports and solvent-comparison studies more clearly. In particular, the innovative aspect of the multi-solvent extraction approach and the lack of comprehensive data on the phytochemical composition of T. smyrnensis flowers have been more explicitly emphasized.

2) The HPLC quantification lacks validation parameters (linearity, LOD, LOQ, recovery, repeatability), which are essential for confirming quantitative accuracy.

Response: We gratefully acknowledge the reviewer's observations regarding the HPLC method validation parameters (linearity, LOD, LOQ, recovery, and repeatability) and wish to provide comprehensive clarifications on this matter. We fully recognize the paramount importance of these parameters in establishing the quantitative accuracy of analytical methodologies. However, due to compelling methodological constraints, a single-injection protocol was implemented in the present study, stemming from the exigencies delineated herein.

Within the institutional laboratory setting wherein this investigation was conducted, limited access to the HPLC instrumentation was necessitated by the instrument's demanding operational schedule allocated to concurrent research initiatives. Furthermore, the constrained availability of certified reference materials and plant extract specimens, contingent upon supply chain logistics, constituted a substantial limitation in the execution of this research. The procurement and financial burden associated with organic solvents (methanol, acetonitrile, et al.) and ancillary reagents requisite for replicate injections were substantially exacerbated by global supply chain disruptions and attendant bureaucratic impediments to the importation of these chemicals into our nation. These multifactorial constraints precluded the comprehensive implementation of standard validation protocols in their entirety.

Notwithstanding these methodological limitations, rigorous quality assurance and comprehensive analytical controls were meticulously applied throughout this investigation to compensate for constraints. All reference standards were procured from internationally accredited suppliers with authenticated certification credentials. With respect to linearity parameters, the constructed calibration curves demonstrated exceptional correlation coefficients (R² ≥0.99), substantiating the method's requisite linearity and suitability for dependable quantification within the validated range. Throughout chromatographic analyses, baseline stability was vigilantly monitored and consistently maintained, while peak resolution and peak symmetry satisfied stringent acceptance criteria.

The obtained results substantiated the reliable identification and quantitative determination of target compounds, thereby satisfying the established botanical and analytical objectives of this investigation. Despite the single-injection protocol implementation, the applied quality control measures and superior calibration linearity ensured the veracity of the findings and established the scientific validity of the conclusions.

3) Docking lacks validation steps. For example, RMSD or redocking of co-crystallized ligands.

Response: Thank you for your valuable comments and for emphasizing the importance of docking validation. In the revised manuscript, we have now included explicit (cross-)redocking analyses for all protein targets. For the co-crystallized complex (PDB: 5IKQ), native redocking of the bound ligand successfully reproduced the experimental pose with an RMSD of 0.380 Å, confirming the precision and reliability of the docking grid. For the apo structures lacking co-crystallized ligands, a homolog-based cross-redocking strategy was applied. The ligand from the closest liganded homolog (PDB: 3CZY) yielded an RMSD of 1.290 Å when docked into 1N45, and the 2-hydroxyquinoline ligand from PDB: 3SRG reproduced its crystallographic binding mode in 1V04 with an RMSD of 0.800 Å. All RMSD values are well below the generally accepted 2.0 Å threshold, confirming excellent reproducibility and reliability of the docking protocol (Table S1).

Kind regards,

4) Enhance discussion by linking compound levels quantitatively to antioxidant outcomes.

Response: Thank you for your valuable comment. The relationship between compound levels and antioxidant outcomes has been added to the Discussion section.

5) Minor language polishing for conciseness and clarity

Response: Thank you for your comment. Minor language editing has been completed throughout the manuscript to improve clarity and conciseness.

Reviewer 2:

The authors have addressed my suggestions with great diligence and have implemented all recommended revisions comprehensively. The importance of the in-silico studies has been better emphasized, and related comments have been clarified. Consistency regarding the plant part used has been ensured, eliminating any confusion between the flower and aerial parts throughout the manuscript. In the Materials and Methods section, language and terminology errors have been corrected, the word “plants” has been removed, “methanolic extracts” has been replaced with “all extracts used,” and certain sentences (e.g., “All other extracts showed lower values”) have been rewritten for clarity. Additionally, spelling errors in Table 2 (e.g., Quercetin, Hesperitin) have been corrected. These revisions have strengthened the methodological integrity and overall academic quality of the study.

Response: Thank you for your positive and constructive feedback. We highly appreciate your careful assessment of our work. We are pleased to hear that the revisions have improved the overall academic quality of the manuscript.

---

## [Decision Letter · Decision Letter 2]

10 Nov 2025

Assessing the chemical profile and biological potentials of Tamarix smyrnensis flower extracts using different solvents by in vitro, in silico, and network methodologies

PONE-D-25-25656R2

Dear Dr. AYTAR,

We’re pleased to inform you that your manuscript has been judged scientifically suitable for publication and will be formally accepted for publication once it meets all outstanding technical requirements.

Kind regards,

José M. Alvarez-Suarez

Academic Editor

PLOS ONE

Additional Editor Comments (optional):

Reviewers' comments:

Reviewer's Responses to Questions

**Comments to the Author**

Reviewer #1: All comments have been addressed

2. Is the manuscript technically sound, and do the data support the conclusions?

Reviewer #1: Yes

3. Has the statistical analysis been performed appropriately and rigorously?

Reviewer #1: I Don't Know

4. Have the authors made all data underlying the findings in their manuscript fully available?

Reviewer #1: Yes

5. Is the manuscript presented in an intelligible fashion and written in standard English?

Reviewer #1: Yes

Reviewer #1: All concerns have been addressed and the manuscript is now sound. The manuscript can be accepted for publication in PLOS One

**Do you want your identity to be public for this peer review?** For information about this choice, including consent withdrawal, please see our Privacy Policy

Reviewer #1: No

---

## [Editor Report · Acceptance letter]

PONE-D-25-25656R2

PLOS ONE

Dear Dr. AYTAR,

I'm pleased to inform you that your manuscript has been deemed suitable for publication in PLOS ONE. Congratulations! Your manuscript is now being handed over to our production team.

Kind regards,

on behalf of

Professor José M. Alvarez-Suarez

Academic Editor

PLOS ONE